# CC-3DT: Panoramic 3D Object Tracking via Cross-Camera Fusion

**Tobias Fischer** *
ETH Zürich
tobias.fischer@vision.ee.ethz.ch

**Yung-Hsu Yang** *
National Tsing Hua University
royyang@gapp.nthu.edu.tw

**Suryansh Kumar**
ETH Zürich
sukumar@vision.ee.ethz.ch

**Min Sun**
National Tsing Hua University
sunmin@ee.nthu.edu.tw

**Fisher Yu**
ETH Zürich
i@yf.io

**Abstract:** To track the 3D locations and trajectories of the other traffic participants at any given time, modern autonomous vehicles are equipped with multiple cameras that cover the vehicle's full surroundings. Yet, camera-based 3D object tracking methods prioritize optimizing the single-camera setup and resort to post-hoc fusion in a multi-camera setup. In this paper, we propose a method for panoramic 3D object tracking, called CC-3DT, that associates and models object trajectories both temporally and across views, and improves the overall tracking consistency. In particular, our method fuses 3D detections from multiple cameras before association, reducing identity switches significantly and improving motion modeling. Our experiments on large-scale driving datasets show that fusion before association leads to a large margin of improvement over post-hoc fusion. We set a new state-of-the-art with 12.6% improvement in average multi-object tracking accuracy (AMOTA) among all camera-based methods on the competitive NuScenes 3D tracking benchmark, outperforming previously published methods by 6.5% in AMOTA with the same 3D detector. Project page: vis.xyz/pub/cc-3dt.

**Keywords:** Autonomous driving, 3D multi-object tracking, Cross-camera fusion

## 1 Introduction

Locating and tracking dynamic objects reliably in 3D is vital to safe autonomous driving systems. Advanced techniques with complex pipelines generally use multiple 3D sensor modalities ranging from stereo cameras to LiDAR and radar. Conversely, a high-quality surround-view camera system provides a low-cost alternative to expensive 3D active sensors to capture a vehicle's surroundings with a panoramic view. Many modern consumer vehicles are equipped with such multi-camera systems. Yet, it is still challenging to effectively utilize temporal and cross-camera association of moving objects in 3D. Such an approach can help to track objects in the long term as cameras can locate them from different viewpoints.

Existing camera-based 3D object tracking methods are designed for single-camera setups [1, 2, 3]. Since those camera-based approaches follow the tracking-by-detection paradigm [4], the major difference between them lies in the association of detections over time [5, 6, 7, 8]. For instance, Zhou et al. [2] relies on 2D motion to associate objects over time, while Chaabane et al. [3] utilizes object appearance. In contrast, Hu et al. [1] combines appearance and 3D motion information. Even though these recent methods perform well on the temporal association of object tracks, they fail to associate object tracks in 3D across multiple cameras installed on the vehicle.

Therefore, we propose an effective *cross-camera fusion* method to the multi-camera 3D tracking problem. We merge 3D detections and their appearance information from multiple cameras into a single 3D tracking component that performs both temporal and cross-camera association. Given

---

*Equal contribution

6th Conference on Robot Learning (CoRL 2022), Auckland, New Zealand.

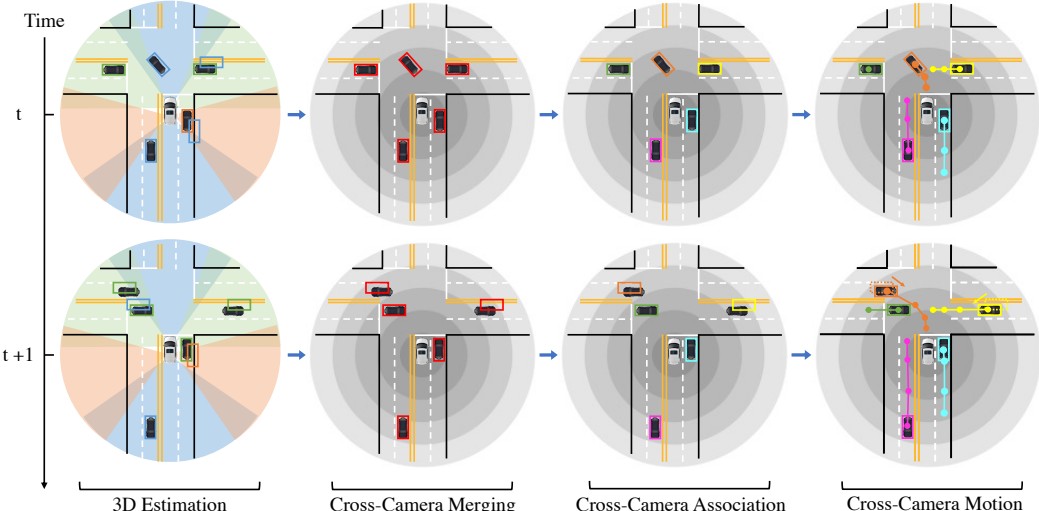

Figure 1: **Method overview**. We first perform 3D object detection and appearance feature extraction for each camera view independently. Then, we lift the camera-space detections into world-space and merge them using non-maximum suppression [11] in 3D. Next, we associate the detections with existing tracks across all cameras. Finally, we refine the detections given the trajectories of the associated tracks across cameras.

the cross-camera association, we leverage a cross-camera motion model to improve the quality of single-frame 3D detections even when an object has just appeared in a particular camera's field of view (FOV), leading to highly accurate tracking with much smoother trajectory of objects in 3D. We call the resulting panoramic 3D object tracking method CC-3DT. The flexible model can work with various 3D object detection methods.

We test CC-3DT on multiple challenging datasets, namely NuScenes [9] and Waymo Open [10]. CC-3DT can associate objects both temporally and across cameras, leading to 17% relative improvement over the single-camera baseline on NuScenes. CC-3DT can exploit the synergy between cross-camera association, leading to longer cross-camera trajectories, and improved object motion modeling, leading to further improved object association. Consequently, we observe significant improvements in both AMOTP and AMOTA scores. Leveraging the cross-camera information not only provides smoother object trajectories but also offers a new state-of-the-art for camera-based 3D object tracking on the competitive NuScenes [9] benchmark, outperforming existing camera-based methods by a large margin.

## 2    Related Work

**Camera-based 3D detection.**    Driven by the success of convolutional neural networks (CNNs) in object detection [11, 12, 13, 14, 15], recent works on camera-based 3D object detection have been using CNNs to regress the 3D parameters of objects from an image [16, 17, 18, 19, 20, 21]. In contrast, another line of works [22, 23, 24] lifts the image to a 3D point cloud using single image depth prediction and detects objects in the 3D point cloud, similar to LiDAR-based 3D object detectors [25, 26, 27]. Lately, transformer networks have been explored for end-to-end object detection [28]. Inspired by this, Wang et al. [29] propose to use a transformer network for 3D object detection from multiple cameras. Our work extends beyond single image 3D detection by performing temporal refinement of the 3D bounding box parameters, improving upon the initial detection result.

**2D multi-object tracking.**    Recent works on MOT follow the tracking-by-detection paradigm [4], where per-frame object detections are associated using different cues, such as 2D motion [30, 5, 31, 32, 33, 2], visual appearance [34, 35, 6, 8, 36, 3, 37, 38] or 3D motion [39, 40, 41, 42, 43]. Recently, researchers have explored the use of transformers in MOT [44, 45, 46]. These works take inspiration from Carion et al. [28] and use the query-based detection process for temporal query propagation to track objects across time with a transformer. In addition, there is a broad range of literature on multi-camera MOT [47, 48, 49, 50, 51]. Those works focus on scenarios with highly

overlapping, stationary surveillance cameras in *e.g.* smart city applications [52]. Their aim is to overcome occlusion problems in monocular MOT systems. On the other hand, our work focuses on multi-camera tracking in the context of autonomous driving, where the cameras usually have little overlap and are mounted on a moving platform.

**Camera-based 3D multi-object tracking.** In the realm of 3D MOT, there are many methods that rely on LiDAR sensors since they provide both accurate 3D localization and 360-degree coverage of the environment [39, 53, 54, 55, 56, 57]. On the contrary, camera-based 3D MOT has been approached by only a handful of works [1, 2, 3]. Those methods are conceptually similar to 2D MOT systems, employing tracking-by-detection using motion or appearance cues for association. While sharing the properties of 2D MOT systems, the camera-based approaches cannot benefit from the 360-degree coverage of the LiDAR-based 3D MOT methods. Nevertheless, none of the methods above consider cross-camera association explicitly. Concurrent to our work, Zhang et al. [58] propose MUTR3D, a method that combines Wang et al. [29] with the ideas in [44, 45, 46] into a transformer-based multi-camera 3D tracking method. However, our work differs significantly from MUTR3D in that we do not rely on a transformer for joint detection and association, but fuse the detections from multiple cameras into a single tracking component. Moreover, our method outperforms MUTR3D significantly in experimental evaluation, even with a weaker 3D detector.

## 3 Method

We first introduce our panoramic 3D tracking pipeline inspired by Hu et al. [1, 59]. Afterward, we detail our cross-camera association and motion modeling. Our method is illustrated in Figure 1.

### 3.1 Overview

Assume we are given a set of input images $I_t$ from cameras $M$ and the corresponding camera locations in world-frame $\xi_t$ in a sequence of time steps $T$. We aim to find the set of tracks $\mathcal{T} = \{\tau^1, \ldots, \tau^N\}$ corresponding to the objects in the scene. Each track $i \in [1, N]$ consists of a set of states $\langle \tau_t^i, \tau_{t+1}^i, \ldots \rangle$ depending on the time steps that the object is visible in. Each state $\tau_t^i = \langle \mathbf{f}_t^{\tau^i}, \mathbf{b}_t^{\tau^i}, c_t^{\tau^i} \rangle$ at a specific time $t \in T$ consists of an appearance embedding $\mathbf{f}_t^{\tau^i} \in \mathbb{R}^{256}$, a 3D bounding box $\mathbf{b}_t^{\tau^i} \in \mathbb{R}^7$ and a confidence value $c_t^{\tau^i} \in [0, 1]$. The 3D bounding box is defined as $\mathbf{b}_t^{\tau^i} = [x, y, z, \theta, l, w, h]$ where $(x, y, z, \theta)$ define the 3D center of the object and its orientation around the vertical axis in 3D world-frame, and $(l, w, h)$ define the object dimensions.

**3D estimation and merging.** We use a 3D estimation network to extract detections $\mathcal{D}_t^m$ from an input image $I_t^m$ of camera $m$ at time $t$. We lift all detections $\mathcal{D}_t^m$ from all cameras $m \in M$ to world space with $\xi_t^m$ and merge them with non-maximum suppression (NMS) in 3D, obtaining multi-camera detections $\mathcal{D}_t$. The detections $d \in \mathcal{D}_t$ share the same state space with tracks $\mathcal{T}$, *i.e.* $d_t = \langle \mathbf{f}_t^d, \mathbf{b}_t^d, c_t^d \rangle$.

**Data association.** We associate the detections $\mathcal{D}_t$ with active tracks $\mathcal{T}_t$ at time $t$ by greedy assignment given an affinity matrix $\mathbf{A}(\mathcal{T}_t, \mathcal{D}_t) \in \mathbb{R}^{||\mathcal{T}_t|| \times ||\mathcal{D}_t||}$ that combines appearance, location, and velocity correlation. We compute the appearance correlation based on the appearance embedding similarity in [37, 38] as

$$\mathbf{a}_{\mathtt{deep}}(\tau_t, d_t) = \frac{1}{2} \left[ \frac{\exp(\mathbf{f}_t^d \cdot \mathbf{f}_t^\tau)}{\sum_{d \in \mathcal{D}} \exp(\mathbf{f}_t^d \cdot \mathbf{f}_t^\tau)} + \frac{\exp(\mathbf{f}_t^d \cdot \mathbf{f}_t^\tau)}{\sum_{\tau \in \mathcal{T}} \exp(\mathbf{f}_t^d \cdot \mathbf{f}_t^\tau)} \right]. \tag{1}$$

The location correlation is based on the absolute difference between the predicted and observed 3D bounding box parameters as

$$\mathbf{a}_{\mathtt{loc}}(\tau_t, d_t) = \exp(\frac{-|\mathbf{b}_t^\tau - \mathbf{b}_t^d|}{r}), \tag{2}$$

where $r$ is a constant scalar. The motion correlation considers the absolute difference between the predicted motion $V_{\tau_t} = \overrightarrow{P_{\tau_{t-1}} P_{\tau_t}}$ and observed pseudo motion $V_{d_t} = \overrightarrow{P_{\tau_{t-1}} P_{d_t}}$ and the centroid distance $|P_{\tau_t} - P_{d_t}|$ as

$$\mathbf{a}_{\mathtt{motion}}(\tau_t, d_t) = w_{\mathtt{cos}} \, \mathbf{a}_{\mathtt{centroid}} + (1 - w_{\mathtt{cos}}) \, \mathbf{a}_{\mathtt{pseudo}}$$

$$\text{where,} \;\; \mathbf{a}_{\mathtt{centroid}} = \exp(\frac{-|P_{\tau_t} - P_{d_t}|}{r}), \; \mathbf{a}_{\mathtt{pseudo}} = \exp(\frac{-|V_{\tau_t} - V_{d_t}|}{r}), \; w_{\mathtt{cos}} = \angle(V_{\tau_t}, V_{d_t}). \tag{3}$$

$w_{\mathtt{cos}}$ is defined as cosine similarity of two motion vectors to favor centroid distance if the motion vectors are in the same hemisphere and pseudo motion otherwise, so that we do not match a pair with conflicting motions. Note that we transform object center $P$ into world-frame using $\xi$ to account for ego-motion. Given the individual affinities, $\mathbf{A}(\mathcal{T}_t, \mathcal{D}_t)$ is computed as

$$\mathbf{A}(\mathcal{T}_t, \mathcal{D}_t) = w_{\mathtt{deep}}\mathbf{A}_{\mathtt{deep}}(\mathcal{T}_t, \mathcal{D}_t) + (1 - w_{\mathtt{deep}})\mathbf{A}_{\mathtt{motion}}(\mathcal{T}_t, \mathcal{D}_t) \cdot \mathbf{A}_{\mathtt{loc}}(\mathcal{T}_t, \mathcal{D}_t), \qquad (4)$$

where $w_{\mathtt{deep}}$ is a constant that weighs appearance and a hybrid of motion and location correlation.

**Motion model.** We employ two LSTM networks $\Phi_{\mathtt{pred}}$ and $\Phi_{\mathtt{update}}$ to handle the extrapolation and update of trajectories given track histories and new observations, respectively. Before data association, we use $\Phi_{\mathtt{pred}}$ to propagate the previous object state $\mathbf{b}_{t-1}$ one time step into the future to compute the location and motion correlation. We input the previous $n - 1 = 5$ object velocities $\mathbf{v}_{t-n:t-1}^{\mathcal{T}}$ and the previous LSTM hidden state $\mathbf{h}_{t-1}^{\mathtt{pred}}$ into the network, where an object velocity is defined as $\mathbf{v}_t^{\mathcal{T}} = \mathbf{b}_t^{\mathcal{T}} - \mathbf{b}_{t-1}^{\mathcal{T}}$. The network outputs the predicted velocity $\hat{\mathbf{v}}_t^{\mathcal{T}}, \mathbf{h}_t^{\mathtt{pred}} = \Phi_{\mathtt{pred}}(\mathbf{v}_{t-n:t-1}^{\mathcal{T}}, \mathbf{h}_{t-1}^{\mathtt{pred}})$. After data association, we use $\Phi_{\mathtt{update}}$ to estimate a velocity refinement given the predicted velocity $\hat{\mathbf{v}}_t^{\mathcal{T}}$, observed velocity $\mathbf{v'}_t^{\mathcal{T}}$, confidence of the associated detection $c_t^{\mathcal{T}}$ and previous hidden state $\mathbf{h}_{t-1}^{\mathtt{pred}}$. The velocity refinement $\mathbf{v}_t^{\mathcal{T}}, \mathbf{h}_t^{\mathtt{update}} = \Phi_{\mathtt{update}}([\hat{\mathbf{v}}_t^{\mathcal{T}}, \mathbf{v'}_t^{\mathcal{T}}, c_t^{\mathcal{T}}], \mathbf{h}_{t-1}^{\mathtt{update}})$ in turn refines the object state $\mathbf{b}_t^{\mathcal{T}} = \mathbf{b}_{t-1}^{\mathcal{T}} + \mathbf{v}_t^{\mathcal{T}}$. Note that for both networks, we first use a linear projection to encode each input into a 64-dimensional vector which we omit for ease of notation.

**Track management.** In order to handle newly appearing, disappearing, and reappearing objects during inference, we keep an online track memory where each track $\tau \in \mathcal{T}$ is assigned one of three possible states $\{\mathtt{active}, \mathtt{inactive}, \mathtt{dead}\}$. At each time step $t$, all tracks currently marked as $\mathtt{active}$ or $\mathtt{inactive}$ are used to calculate their affinity with the current detections $\mathcal{D}_{\sqcup}$. If a detection $d_t \in \mathcal{D}_{\sqcup}$ matches with a track $\tau \in \mathcal{T}, d_t$, it will be assigned to $\tau$. If a track $\tau$ has no matching detection, it will transition into state $\mathtt{inactive}$. If $\tau$ did not have a matching detection for $L$ time steps, $\tau$ transitions into state $\mathtt{dead}$ and will no longer be considered. Note that we propagate the position of inactive tracks forward in time with $\Phi_{\mathtt{pred}}$ to properly match tracks in cases of occlusion. Each detection without matching track spawns a new track in $\mathtt{active}$ state.

### 3.2 Cross-camera association

Hu et al. [1] employs a separate tracker for each camera $m \in M$. In particular, the data association of each camera happens independently, as well as the track management and motion modeling. Since each camera has only a limited FOV, there are many disappearing and re-appearing tracks, making the association prone to error and the trajectories less stable. In addition, object identity will not be preserved across camera views. We explore how a panoramic multi-camera tracker can overcome those limitations with a few simple modifications to the tracking pipeline.

First, we investigate how performance is affected if we use NMS in 3D to naively combine the existing tracks from single-camera trackers. We call this approach detect $\rightarrow$ track $\rightarrow$ merge. Next, we introduce the detect $\rightarrow$ merge $\rightarrow$ track paradigm, *i.e.* instead of aggregating the tracking result of camera-dependent trackers, we *first* aggregate the *detection* results from all cameras $m \in M$ and subsequently apply *global* data association, motion modeling, and track management. The advantages of this are threefold. *(i)* In the regions where the cameras have an overlapping FOV, the multi-camera tracker has access to more detection candidates, meaning that an object that is highly truncated in one camera can be better recognized in a neighboring camera. *(ii)* Cross-camera data association results in long trajectories that span across multiple cameras, which allows for a richer track history and thus better motion modeling. *(iii)* Our data association scheme leverages appearance, location and velocity information to follow tracks more effectively across cameras compared to 3D NMS based merging of single-camera tracking results.

### 3.3 Cross-camera motion modeling

A crucial advantage of our panoramic tracker is that we can exploit longer trajectories across camera views for motion modeling. This leads to an improved multi-frame refinement via $\Phi_{\mathtt{update}}$ and more accurate location prediction via $\Phi_{\mathtt{pred}}$, which in turn benefits the data association since we can expect better estimates for the affinities $\mathbf{a}_{\mathtt{loc}}(\tau_t, d_t)$ and $\mathbf{a}_{\mathtt{motion}}(\tau_t, d_t)$. In addition, we propose to

integrate cross-camera trajectories into the training process of the motion model. The intuition is that due to object truncation, single-view 3D localization at image borders is usually brittle. Thus, if we train the motion model on these trajectories, it can handle those situations better. In contrast to Hu et al. [1], who create the trajectory training data per camera and train on the union of the single-camera trajectories, we create a cross-camera trajectory dataset that matches ground truth trajectories spanning multiple cameras to the aggregated set of detections from all cameras. To match detections to a ground truth trajectory, we use birds-eye-view (BEV) distance instead of Intersection-over-Union (IoU) of 2D bounding boxes in Hu et al. [1].

To train $\Phi_{\texttt{pred}}$ and $\Phi_{\texttt{update}}$, we randomly sample a ground truth trajectory and extract the matched ground truth and detection pairs in a random temporal window $\Delta t$ of fixed size, possibly spanning across multiple camera views. We use the set of detections to simulate inference, *i.e.* we obtain the predicted velocities $\hat{\mathbf{v}}_{\Delta t}$ from $\Phi_{\texttt{pred}}$ given the track history and the refined velocities $\mathbf{v}_{\Delta t}$ from $\Phi_{\texttt{update}}$ given the predicted velocity, the observed velocity, and the detection confidence. We use the ground truth trajectory to calculate the ground truth velocities $\tilde{\mathbf{v}}_{\Delta t}$ and the trajectory loss

$$\mathcal{L}_{\texttt{traj}}(\mathbf{v}_{\Delta t}, \hat{\mathbf{v}}_{\Delta t}, \tilde{\mathbf{v}}_{\Delta t}) = \frac{1}{||\Delta t||} \sum_{t \in \Delta t} \delta(\hat{\mathbf{v}}_t, \tilde{\mathbf{v}}_t) + \delta(\mathbf{v}_t, \tilde{\mathbf{v}}_t), \tag{5}$$

where $\delta(\cdot, \cdot)$ is the Huber loss [60]. Moreover, we add a regularization term that encourages $\Phi_{\texttt{pred}}$ to predict a smooth trajectory over $\Delta t$:

$$\mathcal{L}_{\texttt{linear}}(\hat{\mathbf{v}}_{\Delta t}) = \frac{1}{||\Delta t||} \sum_{t \in \Delta t} ||(\hat{\mathbf{v}}_{t+1} - \hat{\mathbf{v}}_t) - (\hat{\mathbf{v}}_t - \hat{\mathbf{v}}_{t-1})||_1. \tag{6}$$

Finally, we calculate the overall loss $\mathcal{L}_{\texttt{motion}}$ as a weighted combination of the above terms

$$\mathcal{L}_{\texttt{motion}}(\mathbf{v}_{\Delta t}, \hat{\mathbf{v}}_{\Delta t}, \tilde{\mathbf{v}}_{\Delta t}) = \mathcal{L}_{\texttt{traj}}(\mathbf{v}_{\Delta t}, \hat{\mathbf{v}}_{\Delta t}, \tilde{\mathbf{v}}_{\Delta t}) + w_{\texttt{linear}} \mathcal{L}_{\texttt{linear}}(\hat{\mathbf{v}}_{\Delta t}). \tag{7}$$

## 4 Experiments

**Datasets.** We evaluate our method on two popular 3D MOT benchmarks. First, we experiment on the NuScenes benchmark [9] which consists of street scenes captured in varying scenery and geographic areas in Boston and Singapore. The scenes are captured from a moving vehicle with a multiple sensors attached. It provides six different camera views with a 360-degree coverage of the environment. In total, the dataset contains 1000 sequences sampled at 2 Hz. Second, we evaluate our method on the Waymo Open benchmark [10], which has similar characteristics to the NuScenes benchmark. However, the sampling rate is 10 Hz, and the five cameras mounted on the vehicle cover only about a $5/8$ azimuth angle of the horizon.

**Evaluation metrics.** We follow the evaluation protocol in NuScenes using the metrics in Weng et al. [53]. The main metrics are average multi object tracking accuracy (AMOTA) and average multi object tracking precision (AMOTP) which are integrals over MOTA / MOTP [61] scores computed via n-point interpolation. AMOTA is defined as $\text{AMOTA} = \frac{1}{n-1} \sum_{r \in \{\frac{1}{n-1}, \frac{2}{n-1}, ..., 1\}} \text{MOTA}_r$, where $\text{MOTA}_r = \max\left(0, 1 - \frac{\text{IDS}_r + \text{FP}_r + \text{FN}_r + -(1-r)\text{P}}{r\text{P}}\right)$. Specifically, $\text{MOTA}_r$ is a recall normalized MOTA and $\text{IDS}_r$, $\text{FP}_r$, $\text{FN}_r$ are number of ID switches, false positives and false negatives at recall $r$. P is the total number of ground truth positives. AMOTP is defined as $\text{AMOTP} = \frac{1}{n-1} \sum_{r \in \{\frac{1}{n-1}, \frac{2}{n-1}, ..., 1\}} \frac{\sum_{i,t} d_{i,t}}{\sum_t \text{TP}_t}$ where $d_{i,t}$ is the BEV distance between track $i$ and its ground truth track at time $t$ and $TP_t$ denotes the number of matched ground truth tracks at time $t$.

**Implementation details.** We train the 3D estimation network using ResNet-101 [62] as backbone with batch size of 32 for 24 epochs with random image flipping as data augmentation on 8 V100 GPUs. The initial learning rate is set to 0.01 with a linear learning rate warm-up in the first 1000 iterations. We decay the learning rate by a factor of 10 at epochs 16 and 22, respectively. We use SGD [63] with a momentum of 0.9 and a weight decay of 0.0001. For NuScenes, we use $1600 \times 900$ resolution and train on the training set only. On Waymo Open, we scale the images to $1280 \times 854$ resolution and use batch size of 16 during training. We decay the learning rate by a factor of 5 at epochs $8, 12, 16, 20, 22$, respectively. To merge 3D detections across cameras, we use NMS with a 3D IoU threshold of 0.1. To generate the cross-camera trajectory dataset, we use a matching threshold of 2 meters BEV distance. We train $\Phi_{\texttt{pred}}$ and $\Phi_{\texttt{update}}$ with batch size of 128 for 100 epochs. We use $w_{\texttt{linear}} = 0.001$ and Adam [64] with AMSGrad and a weight decay of 0.0001.

Table 1: **Pipeline and motion modeling ablation study.** We compare the tracking performance of our method with the single-camera baseline [1] and the detect → track → merge paradigm on the NuScenes validation split. We further analyze the benefit of motion modeling across all pipelines, both with LSTM motion model and a Kalman Filter baseline (KF3D).

| Pipeline | Motion Model | AMOTA ↑ | AMOTP ↓ | RECALL ↑ | MOTA ↑ | IDS ↓ |
|---|---|---|---|---|---|---|
| QD-3DT [1] | - | 0.221 | 1.529 | 0.371 | 0.193 | 5256 |
| | KF3D | 0.232 | 1.530 | 0.393 | 0.206 | 5574 |
| | LSTM | 0.247 | 1.507 | 0.399 | 0.218 | 5919 |
| Detect → Track → Merge | - | 0.248 | 1.500 | 0.386 | 0.212 | 4451 |
| | KF3D | 0.256 | 1.506 | 0.408 | 0.225 | 4256 |
| | LSTM | 0.264 | **1.490** | 0.413 | 0.235 | 4470 |
| Detect → Merge → Track | - | 0.249 | **1.490** | 0.405 | 0.212 | 2399 |
| | KF3D | 0.279 | 1.501 | 0.415 | 0.252 | **1982** |
| | LSTM | **0.283** | **1.490** | **0.444** | **0.262** | 2131 |

Table 2: **Motion model training ablation study.** We compare our modified training scheme for the LSTM motion model with cross-camera trajectories to the single-camera baseline on the NuScenes validation split. † denotes using new training scheme for 3D estimation network.

| Method | Trajectory data | matching | Total Trajectories | AMOTA ↑ | AMOTP ↓ | IDS ↓ |
|---|---|---|---|---|---|---|
| CC-3DT | Single-camera | 2D IoU | 29527 | 0.283 | 1.490 | 2131 |
| CC-3DT | Cross-camera | BEV Distance | 36112 | 0.289 | 1.483 | **2129** |
| CC-3DT† | | | | **0.311** | **1.433** | 2536 |

## 4.1 Ablation studies

**Cross-camera association.** In Table 1, we compare the pipeline in Hu et al. [1], naive merging of single camera results (detect → track → merge) and our pipeline described in section 3.2 using the LSTM motion model, a Kalman Filter baseline (KF3D) and no motion model. Note that all individual components are consistent while only changing the high-level pipeline to provide a fair comparison. We observe that naive merging helps the overall performance and results in a gain of 1.7% in AMOTA using the LSTM motion model. With our cross-camera association (detect → merge → track), we observe another 1.9% gain in AMOTA while the IDS are cut in half, highlighting the effectiveness of cross-camera association. Note that while KF3D has slightly less IDS than LSTM, its recall level is also lower, *i.e.* IDS are lower due to lower recall not better association. Interestingly, the gain of cross-camera association is very small when not using a motion model (0.1%), supporting our hypothesis that the motion model benefits greatly from cross-camera association. Moreover, the influence of the motion model on the final tracking performance is the highest when using cross-camera association, improving from 24.9% to 28.3% AMOTA. Our results underline the importance of cross-camera association to achieve consistent tracking results.

**Cross-camera motion modeling.** Table 1 also shows that motion modeling improves through the use of cross-camera trajectories, *i.e.* we observe a higher gain in AMOTA of 3.4% (LSTM) and 2.0% (KF3D) versus using no motion model compared to only 2.6% (LSTM) and 1.1% (KF3D) when using the pipeline in Hu et al. [1]. The effect of training the motion model with cross-camera trajectories (see section 3.3) is illustrated in Table 2. We observe another improvement of 0.6% in AMOTA compared to utilizing cross-camera trajectories only during inference. Moreover, Table 2 shows that the total number of trajectories for training increases, which can be attributed to the fact that for a trajectory to be considered for training, it needs to be at least $||\Delta t||$ time steps long. Cross-camera trajectories help to increase training data diversity. In sum, we achieve a gain of 4.2% in AMOTA, equating to a 17% relative improvement compared to the single-camera baseline.

**Qualitative comparison.** Figure 2 illustrates CC-3DT results compared to the baselines in Table 1. While our proposed CC-3DT preserves object identity and exhibits a smooth, accurate trajectory for the object moving across camera views, the baselines struggle with object localization and object association. Specifically, QD-3DT [1] cannot preserve the object identity while naive merging (detect → track → merge) results in poor localization, showing the benefit of our method.

Table 3: **Comparison to MUTR3D [58] using different motion models**. We report tracking performance on the NuScenes validation split. We use DETR3D [29] as detector for fair comparison.

| Method | Motion model | AMOTA ↑ | AMOTP ↓ | RECALL ↑ | MOTA ↑ | IDS ↓ |
|---|---|---|---|---|---|---|
| MUTR3D [58] | SimpleTrack | 0.293 | **1.307** | 0.418 | 0.263 | **1695** |
| | FFN | 0.294 | 1.498 | 0.427 | 0.267 | 3822 |
| CC-3DT | - | 0.307 | 1.394 | 0.434 | 0.263 | 2419 |
| | KF3D | 0.353 | 1.379 | 0.478 | 0.316 | 2050 |
| | LSTM | **0.359** | 1.361 | **0.498** | **0.326** | 2152 |

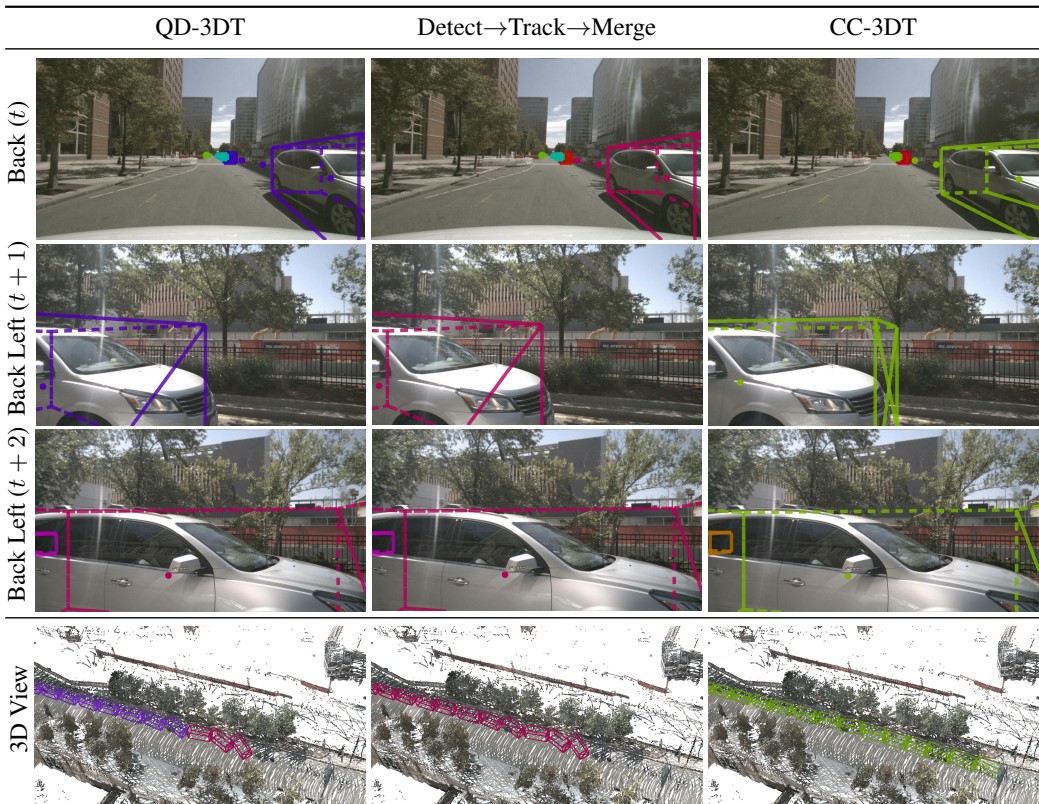

Figure 2: **Qualitative comparison.** Different types of cross-camera association on NuScenes validation split. We plot the 3D car states in camera and 3D views and depict car identity as color. Clearly, CC-3DT provides a smooth and precise trajectory across cameras (green). Detect → Track → Merge shows awry car localization (purple), while QD-3DT lacks car identity (violet).

**3D estimation network training scheme.** Following [29] and different from [1], we optimize the 3D estimation network training scheme by increasing the learning rate of the 3D detection head by a factor of 10. As shown in Table 2, this improves the overall performance by 2.2% in AMOTA. Note that IDS increase due to increased recall of our detector.

## 4.2 Comparison to state-of-the-art

**Comparison on the NuScenes benchmark.** We evaluate our method the NuScenes benchmark in Table 4. On the validation set, we outperform the state-of-the-art method MUTR3D by 1.7% with a weaker detector, by 6.5% in AMOTA with the same detector (DETR3D [29]), and by 13.5% with a stronger detector (BEVFormer [66]). Our AMOTP is significantly lower (1.361 vs. 1.498) with the same detector, meaning that our trajectories are more precise. We also have the lowest IDS, meaning that our association works best. The testing set results corroborate our findings, *i.e.* we outperform MUTR3D by as much as 12.4% in AMOTA and achieve the lowest AMOTP and IDS. We set a new state-of-the-art of 42.9% and 41.0% AMOTA on the validation and testing sets, respectively.

Table 4: **Comparison to state-of-the-art.** We compare the tracking performance of our method with existing camera-based methods on the NuScenes validation and testing sets.

| | Method | Detector | AMOTA ↑ | AMOTP ↓ | RECALL ↑ | MOTA ↑ | IDS ↓ |
|---|---|---|---|---|---|---|---|
| **Validation** | DEFT [3] | CenterNet [65] | 0.201 | N/A | N/A | 0.171 | N/A |
| | QD-3DT [1] | Faster-RCNN [11] | 0.247 | 1.507 | 0.405 | 0.221 | 5919 |
| | MUTR3D [58] | DETR3D [29] | 0.294 | 1.498 | 0.427 | 0.267 | 3822 |
| | CC-3DT | Faster-RCNN [11] | 0.311 | 1.433 | 0.472 | 0.278 | 2536 |
| | | DETR3D [29] | 0.359 | 1.361 | 0.498 | 0.326 | **2152** |
| | | BEVFormer [66] | **0.429** | **1.257** | **0.534** | **0.385** | 2219 |
| **Testing** | CenterTrack [2] | CenterNet [65] | 0.046 | 1.543 | 0.233 | 0.043 | 3807 |
| | DEFT [3] | CenterNet [65] | 0.177 | 1.564 | 0.338 | 0.156 | 6901 |
| | QD-3DT [1] | Faster-RCNN [11] | 0.217 | 1.550 | 0.375 | 0.198 | 6856 |
| | MUTR3D [58] | DETR3D [29] | 0.270 | 1.494 | 0.411 | 0.245 | 6018 |
| | CC-3DT | Faster-RCNN [11] | 0.284 | 1.470 | 0.427 | 0.251 | **2945** |
| | | DETR3D [29] | 0.357 | 1.343 | 0.472 | 0.317 | 3070 |
| | | BEVFormer [66] | **0.410** | **1.274** | **0.538** | **0.357** | 3334 |

**Comparison to MUTR3D.** Table 3 provides additional comparison to MUTR3D using different motion models. MUTR3D reports results with a Kalman Filter baseline [67] and a feed-forward network (FFN). While the FFN can only improve marginally upon the Kalman Filter baseline, our motion model significantly outperforms the Kalman Filter baseline (KF3D). Also, our cross-camera association is more effective than the transformer-based one in MUTR3D (lower IDS). Note that while MUTR3D with Kalman Filter has the lowest IDS, the detection recall is also much lower.

**Comparison on the Waymo Open benchmark.** We compare with [1] on the validation split of Waymo Open. The comparison in Table 5 shows that our method performs better in MOTA than QD-3DT, while also having a lower Mismatch ratio, meaning that we have both better overall and association performance. We outperform QD-3DT significantly on each of the 3D IoU thresholds. Note that we do not evaluate with an IoU threshold of 0.7, as it is too restrictive for camera-based methods to achieve a meaningful score.

Table 5: **Comparison on Waymo Open.** We compare our 3D Vehicle Tracking results with QD-3DT on the validation set of the Waymo Open benchmark (Level 2 difficulty).

| Method | 3D IoU | MOTA ↑ | Mismatch ↓ |
|---|---|---|---|
| QD-3DT [1] | 0.3 | 0.1867 | 0.0134 |
| | 0.5 | 0.0308 | 0.0055 |
| CC-3DT | 0.3 | **0.2032** | **0.0069** |
| | 0.5 | **0.0480** | **0.0018** |

### 4.3  Limitations

Our method could fail if the predicted 3D detection is amiss. Specifically, in cases of high occlusion or truncation, the object's 3D properties are often ambiguous to the observer, *e.g.* an object's 3D center could even be out of the camera FOV. This is especially pronounced for big or close objects that are often only partially visible. While combining information from multiple time steps in our motion model alleviates those issues, integrating this information during the detection stage could further improve the results. These challenges can also influence the classification performance of the 3D detector. If an object is highly occluded or truncated, it is often challenging to infer its category from a single image, especially for semantically similar classes (*e.g.* bus and truck). However, if the category is estimated incorrectly, object identity may not be preserved.

## 5  Conclusion

This paper presents an effective 3D object tracking method in a multi-camera setting. The key idea we present is to merge 3D detections of multiple cameras into a single 3D tracking component that handles both temporal and cross-camera association. By leveraging cross-camera association, we observe improved motion model quality, leading to enhanced object association and smoother trajectories across camera views. As a result, the suggested approach outperforms existing methods by a large margin achieving state-of-the-art results on the widely-used NuScenes benchmark.

# 6 Acknowledgments

This work is supported by Ministry of Science and Technology of Taiwan (MOST 110-2634-F-002-051). We gratefully thank National Center for High-performance (NCHC) for the storage and computational resource.

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
