# OpenReview forum: "CC-3DT: Panoramic 3D Object Tracking via Cross-Camera Fusion"
_robot-learning.org/CoRL/2022/Conference — CoRL 2022 Poster_

### Official Review · Reviewer_ANzg · 2022-07-27

**Originality:** Very Good
**Technical Quality:** Very Good
**Clarity Of Presentation:** Very Good
**Impact:** 4

**Recommendation:**

Weak Accept: I recommend accepting the paper, but will not argue for my recommendation if the majority of other reviewers have a different opinion.

**Summary:**

The paper “MV-3DT: Multi-View 3D Object Tracking via Cross-Camera 3D Fusion” is proposing a pipeline for tracking objects across multiple cameras and time for autonomous driving applications. The pipeline is relying on a monocular 3D object detection network that predicts bounding boxes for each image in a multi camera system. Afterwards, the object detections are associated across all cameras and merged using 3D non-maximum suppression. The merged 3D objects are tracked over time using a learned LSTM-based motion model. Furthermore, a track management is proposed, where each object track can have the following states: active, inactive, dead. This way objects can be tracked even if they are not detected over a longer time frame.
The proposed pipeline is evaluated against recent single-camera (QD-3DT) and multi-camera (MURT3D) on the NuScenes and Waymo Open dataset. On both datasets the proposed method is outperforming the baselines.

**Issues:**

* Add runtime measurements for the pipeline
* Move qualitative results from supplementary to main paper

**Quality Of The Limitations Section:**

Limitations are addressed clearly

**Reviewer Expertise:**

4: The reviewer is confident but not absolutely certain that the evaluation is correct

**Robotics Focus:**

Highly relevant to robotics but no hardware experiments

**Strengths And Weaknesses:**

Strengths:
* The paper is very well structured and easy to read. Therefore, one can easily get the idea of the proposed method and how it differentiates from the existing state of the art
* All design choice are well explain and a validated in the presented ablation studies
* Implementation details are given which can be helpful to reproduce the results presented in the paper
* Extensive experiments are perform to evaluate the proposed pipeline against existing state-of-the-art approaches

Weaknesses:
* The presented paper shows only minor weaknesses, which are as follows:
It would be beneficial to also report runtimes in the evaluations section. E.g. it is shown that the LSTM-based motion model is outperforming a classical EKF. However, it would be beneficial to understand how both model differ in runtime.
* To improve the presentation of the paper it would be beneficial to move some of the qualitative results from the supplementary to the main paper.

**Summary Of Recommendation:**

The paper is nicely written and easy to follow. Furthermore, it is proposing a novel approach and clearly explaining the improvements to the current state-of-the-art. The performance is evaluated based on extensive experiments.

---

> ### Author Response · Authors · 2022-08-24
> **We thank you for the thoughtful feedback.**
>
> We thank you for the thoughtful feedback. Following your comments, we added a runtime analysis in the supplementary material. Overall, the majority of the execution time is occupied by the 3D detector, which our method is agnostic of (we present results with 3 different detection methods). In comparison, the LSTM networks are quite small and only occupy a small amount of the total runtime. However, the KF3D baseline is indeed faster. Furthermore, we revised figure 2 in the main paper to align it with the qualitative results in the supplementary material. We hope that this settles the issues you mentioned.

---

### Official Review · Reviewer_qceo · 2022-07-31

**Originality:** Good
**Technical Quality:** Very Good
**Clarity Of Presentation:** Good
**Impact:** 3

**Recommendation:**

Weak Accept: I recommend accepting the paper, but will not argue for my recommendation if the majority of other reviewers have a different opinion.

**Summary:**

This paper presents a new way targeting at 3D object tracking in a multi-camera setup. Compared to previous post-hoc fusion approaches, this paper presents presents two main contributions:

1. cross-camera ssociation;
2. cross-camera motion modelling.

The idea of fusing 3D detections from individual camera inputs before data association and then predicting and tracking objects is throughly verified in two different benchmark datasets, showing the tracking accuracy improvement brought by the contributions in the paper


**Issues:**

First of all, I would like to admit that I do not have a lot of experience in the field of 3D MOT. From my perspective, the idea of fusing information across camera views before tracking is a good idea. However, after reading this paper, I have a few questions in my head.
1. The whole paper presents a large system, with different components using different network components. For better reproduction of this paper, it will be better to explain the architecture of each component, for example, in the supplementary materials.
2. There are some unexplained hyparameters within each model, such as the number of object velocities (L119), embedding dimension (L93), deep weight $w_deep$ (L115), and track management window $L$ (L134).Those parameters probably affect the performance of the system. It is not explained in the main paper or the supplementary document what values are used for these parameters in the experiments and how they are tuned.
3. Why does Eq (4) choose to directly multiply the motion and velocity correnlation matrix?
4. When referring to the main contribution in section 3.1 and 3.2, this paper frequently refers to [6], leaving an impression that [6] serves as a baseline of its system. However, system-wise, it is not clearly explained which components are inspired or borrowed from [6].
5. As a submission to a robotics conference, I would expect to see some analysis (ideally a break-down of the components) of the computation time of the proposed method. For example, compared to the Kalman filter, how much of a computation time trade off does it make to gain the accuracy improvement?
6. The experiments were conducted on two benchmarks. In the implementation details, it seems that different hyperparameters are needed to tune for different datasets. From my perspective, it is fine, but an explanation may be required. I assume the same training setting does not work well. Is it because of the sampling rate and the FOV difference or something else? And does it mean that the trained model may not generalise well to different sensor setups?

**Quality Of The Limitations Section:**

Additional details required

**Reviewer Expertise:**

3: The reviewer is fairly confident that the evaluation is correct

**Robotics Focus:**

Highly relevant to robotics but no hardware experiments

**Strengths And Weaknesses:**

Strengths:
The idea of fusing the detections before performing tracking and predicting object trajectories is good and has been throughly verified in the experiment section.

Weakness:
This paper presents a quite large system, with many components and hyparameters. To understand the core contribution of this paper better, it will be better to address the issues that I have raised below.

**Summary Of Recommendation:**

This paper shows an advancement in the multi-camera 3D MOT field. The idea is not difficult but seems to work well. There are a few points that could be further explained better.

---

> ### Author Response · Authors · 2022-08-24
> **We thank you for your review and appreciate your questions. Below we hope to address them**
>
> We thank you for your review and appreciate your questions. Below we hope to address them.
>
> Q1 + 2: Architecture of components and details on hyperparameters
>
> We describe the architecture of the different components and discuss the hyperparameter values of our 3D tracker in the updated supplemental material.
>
> Q3: Multiplication of motion and location affinity matrices
>
> We addressed this question in the answer to Reviewer tUyR (kindly refer Q1).
>
> Q4: Comparison to [6]
>
> We adopt the 3D estimation network and the LSTM networks from [6]. Our contribution lies in proposing a multi-camera tracking workflow (‘cross-camera fusion’) and a multi-camera training scheme for the LSTM motion model, meaning that we change the high-level pipeline, while keeping the internal components consistent to make it suitable for detailed ablation study in table 1. We hope the revised manuscript makes this more clear.
>
> Q5: Runtime comparison
>
> We added a runtime comparison to the supplemental material. Overall, the majority of the execution time is occupied by the 3D detector, which our method is agnostic of (we present results with 3 different detection methods). In comparison, the LSTM networks are quite small and only occupy a small amount of the total runtime. However, the KF3D baseline is indeed faster.
>
> Q6: Different hyperparameters and generalization across datasets
>
> Indeed, the 3D detection model is trained on each dataset separately, and as those have different sensor setups, the training protocol differs slightly to achieve best performance. However, the tracking pipeline after is, as mentioned above, agnostic of the 3D detector and thus generalizes better across datasets.

---

> > ### Comment · Reviewer_qceo · 2022-08-24
> > **Response to authors**
> >
> > I would like to thank the authors for answering my questions and updating the related information in the manuscripts. Considering all the reviews and the authors' responses, I am leaning towards keeping my rating.

---

### Official Review · Reviewer_tUyR · 2022-07-31

**Originality:** Very Good
**Technical Quality:** Good
**Clarity Of Presentation:** Good
**Impact:** 4

**Recommendation:**

Weak Accept: I recommend accepting the paper, but will not argue for my recommendation if the majority of other reviewers have a different opinion.

**Summary:**

Accurate and consistent object tracking across multiple cameras is a critical necessity for autonomous vehicles to safely navigate in the real world. The authors introduce a cross-camera 3d fusion method, called MV-3DT, that fuses 3D object detection results from multiple cameras before object identity association while tracking objects in a multi-camera setup. In contrast, existing camera-based 3D object tracking methods first detect objects using a single-camera setup and later post-hoc fuse detections to track objects in a multi-camera setup. Experiments conducted by the authors on large-scale driving datasets namely NuScenes 3D tracking benchmark and Waymo open benchmark empirically demonstrate that by fusing 3D detection results before object association, MV-3DT results in better average multi-object tracking accuracy compared to the existing methods with a significant reduction in object identity switches temporally and across cameras.

**Issues:**

- The symbol $I^m_t$ has been used twice (in lines 89 and 98) to denote two different entities, first given input images from a set of cameras and second, a single input image to the 3D estimation network. Different symbols should be used to denote different entities.
- The affinity matrix A is computed as a weighted sum of the appearance affinity and a hybrid affinity of motion and location correlation. Please clarify why a hybrid affinity of motion and location correlation was considered instead of adding them as individual affinities.
- In Table 1, the baseline KF3D leads to fewer identity switches compared to the proposed VeloLSTM method (see the last couple of rows). Similarly, in Table 2, MV-3DT without the new training scheme for the 3D estimation network reports fewer object identity switches compared to when the new training scheme is used. Please discuss potential reasons for these in the text and how to mitigate them.
- In Figure 2, it is difficult to follow what the major implications of the plots are. For example, it is not clear what the green, yellow and red circles represent. If red circles represent smooth and precise object tracking, then why are there no red circles around the blue and green objects in the first row plots?
- Briefly describe what are the major implications of the data presented in the tables in the caption. This helps the reader to quickly read and appreciate the experimental results.

Minor comments:
- In line 80, please say that “... Zhang et al. [60] propose MUTR3D, a method ….” as it is an important baseline and mentioning its name here helps the reader to follow the discussion more clearly.
- Rephrase statement “... i.e. 2.6% vs. 1.6% vs. 3.4% difference 223 of using VeloLSTM versus using no motion model.” in lines 223-224 for better readability.


**Quality Of The Limitations Section:**

Limitations are addressed clearly

**Reviewer Expertise:**

3: The reviewer is fairly confident that the evaluation is correct

**Robotics Focus:**

Highly relevant to robotics but no hardware experiments

**Strengths And Weaknesses:**

Strengths:
- The authors present an intuitive and effective idea of how information across multiple cameras should be combined to have accurate and consistent object tracking.
- The manuscript is well written and captures the main ideas of the work comprehensively. Showing tracked objects across cameras in figure 1 is quite informative.
- Videos showcasing object tracking accuracy across multiple cameras were quite insightful in appreciating the effectiveness of the work.

Weaknesses:
- Bird-eye view figures, especially figure 2, can be improved to make it easier for the reader to follow tracked objects across time steps. Discussed further in the “Issues” section.
- Further quantitative results on the Waymo open benchmark should also be included.


**Summary Of Recommendation:**

The ideas introduced in the work are novel and exciting. The manuscript is well-written. It presents the core ideas comprehensively, warranting an acceptance of the work at the conference. However, the authors must address a few issues before I would be confident in recommending the work for publication. If the authors address the raised concerns in their rebuttal satisfactorily, I would be happy to recommend the manuscript for publication.

---

> ### Author Response · Authors · 2022-08-24
> **We thank you and appreciate the thorough review and the comments and questions. Below we hope to address your suggestions**
>
> We thank you and appreciate the thorough review and the comments and questions. Below we hope to address your suggestions.
>
> Q1: Why use a hybrid of motion and location correlation
>
> We use this formulation because we want to constrain the affinity score to emphasize consistent 3D motion and location, i.e. we don’t want a high affinity when a track / detection pair has consistent motion but inconsistent location and vice versa (similar to a logical and relation). On the other hand, appearance is independent of both location and motion and thus we use a weighted summation (similar to a logical or relation).
>
> Q2: IDS KF3D vs LSTM
>
> The reason KF3D reports less IDS than VeloLSTM is similar for both of the tables. In Table 1, we are able to preserve more trajectories with the refinement of the LSTM motion model (having more true positives), as can be seen by the higher recall value. However, the more recall you have, the more IDS you will have because the IDS is given as an absolute number in the NuScenes benchmark, not as a percentage. The same applies for Table 2, where our 3D detector improves (higher recall) through the re-training, and thus our IDS rises. To be clear, we still have the same error rate, however the more trajectories you have, the higher the absolute number of errors. We agree that this relation is not obvious, and added explanations in the text.
>
> C1: Qualitative results in figure 2.
>
> We completely revised the presentation of figure 2. We hope that our new illustration captures the advantages of our work.
>
> C2: Additional quantitative results on Waymo
>
> We appreciate the suggestion, however we included the full comparison on the Waymo benchmark to the results in QD-3DT [6], which is the only camera-based 3D MOT method to report quantitative results on the Waymo benchmark. If you could point us to another work that reports more quantitative results on Waymo, we would be happy to include it in the comparison.
>
> C5: Describe major implications of results in Table captions.
>
> Thanks a lot for the suggestion, we have revised all Table captions. We hope it improves the readability of the paper.
>
> Finally, we addressed the writing style specific comments in the updated version of our manuscript, thank you for your constructive feedback.

---

### Official Review · Reviewer_qyz1 · 2022-08-01

**Originality:** Fair
**Technical Quality:** Good
**Clarity Of Presentation:** Good
**Impact:** 3

**Recommendation:**

Weak Reject: I recommend rejecting the paper, but will not argue for my recommendation if the majority of other reviewers have a different opinion.

**Summary:**

This paper proposes MV-3DT, a multi-camera 3D object tracking pipeline. The major novelty of MV-3DT is how it handles detections results from multiple cameras, where unlike existing works, it aggerates all cameras' detections together and performs temporal track matching across all cameras. MV-3DT performances on large-scale real-world datasets are comparable with base models and is STOA on NuScenes. Also, the paper includes limited ablation study to demonstrate the performance contribution of their proposed multi-camera association pipeline.

**Issues:**

1. For 3D estimation network, the actual model architecture detail as well as the training loss for its three heads are not mentioned in the paper.
2. For Cross-camera association, the merge state detail is missing, the paper doesn't discuss how tracks or detections are aggregated. For example for the later detect -> merge -> track pipeline, how is detections of the same object appears in two difference cameras be aggregated?
3. For section 3.2, the paper claims "The intuition is that due to object truncation, the 3D localization at image borders is usually brittle", and uses this as support for the idea of cross-camera trajectories. However, my argument is that merging multi-cameras images into a single panoramic image could handle the image border problem even better.  As a follow-up question about this paper's comparison result is that, there is no comparison with methods using bird eye view based detection methods, and I think that is an important part to justify the proposed model is better.

**Quality Of The Limitations Section:**

Limitations are addressed clearly

**Reviewer Expertise:**

4: The reviewer is confident but not absolutely certain that the evaluation is correct

**Robotics Focus:**

Highly relevant to robotics but no hardware experiments

**Strengths And Weaknesses:**

Strengths:
1. This paper demonstrates MV-3DT performances across multiple large scale datasets and compare it with many base models.
2.  The idea of using two LSTM to handle loss-tracked objects and tracking objects separately is an interesting approach for disappearing and reappearing objects.
Weakness
1. For 3D estimation network, the actual model architecture detail as well as the training loss for its three heads are not mentioned in the paper.
2. For Cross-camera association, the merge state detail is missing, the paper doesn't discuss how tracks or detections are aggregated. For example for the later detect -> merge -> track pipeline, how are detections of the same object appears in two different cameras be aggregated?
3. For section 3.2, the paper claims "The intuition is that due to object truncation, the 3D localization at image borders is usually brittle", and uses this as support for the idea of cross-camera trajectories. However, my argument is that merging multi-cameras images into a single panoramic image could handle the image border problem even better.  As a follow-up question about this paper's comparison result is that, there is no comparison with methods using bird eye view based detection methods, and I think that is an important part to justify the proposed model is better.
4. The novelty of this paper is weak, merge detection before track doesn't seem to be enough to justify the novelty

**Summary Of Recommendation:**

This paper proposes MV-3DT, a multi-camera 3D object tracking pipeline. The major novelty of MV-3DT is how it handles detections results from multiple cameras, where unlike existing works, it aggerates all cameras' detections together and performs temporal track matching across all cameras. MV-3DT performances on large-scale real-world datasets are comparable with base models and is STOA on NuScenes.
The novelty of this paper is weak, merge detection before track doesn't seem to be enough to justify the novelty. Some model details and pipeline processing methods are not clearly stated in the paper. Also, the paper includes limited ablation study to demonstrate the performance contribution of their proposed multi-camera association pipeline, yet the claim of "reduces identity switches significantly" is not supported by ablation study.

---

> ### Author Response · Authors · 2022-08-24
> **Thank you for the feedback and comments on our paper. Through our response, we hope to address your questions and comments.**
>
> Thank you for the feedback and comments on our paper. Through our response, we hope to address your questions and comments.
>
> Q1: 3D estimation network architecture and losses
>
> We provide details on the 3D estimation network in the supplemental material. We use a standard Faster-RCNN architecture with ResNet101-FPN backbone and the same 3D detection/appearance heads as in QD-3DT. Hence, the 3D detection and appearance embedding losses are consistent with this work. However, our method is agnostic of the 3D detection method, and we provide results with multiple different 3D detection models.
>
> Q2: Details on cross-camera association
>
> We revised section 3 (see 3.1 "overview") and the overview illustration in figure 1 to make it more clear. Briefly, we aggregate the detections by lifting them into world space and using non-maximum suppression in 3D. We hope our revised draft addresses this concern.
>
>
> C1: Cross-camera trajectories and comparison when using BEV detection methods
>
> A multi-camera 3D detection system could be a better choice for this task, and therefore we include results with the multi-camera 3D detection method DETR3D.  In addition, we included results with BEVFormer [Li et al. 2022] as 3D detector in our supplementary material. Following your suggestion, we moved those results to the main paper. We observe that for both of those multi-camera detection systems, we improve substantially upon the state-of-the-art (MUTR3D), even when using the same detector (DETR3D). We hope that this highlights the importance of our cross-camera association and trajectories, independent of the 3D detection method at hand.
>
> C2: Justification of novelty w.r.t. cross-camera fusion
>
> Contrary to the existing methods [6-8], which differ only in the affinities they use for the correlation matrix, our paper presents an algorithmic novelty by explicitly handling cross-camera association and motion modeling. Hence, our pipeline is indeed more novel than proposing a new affinity for 3D MOT, which is the main contribution of the previous works. While merging detections before tracking is a simple and intuitive idea, our experimental evaluation shows how crucial it is to performance. Furthermore, to the best of our knowledge, we are not aware of any existing work that proposes such an idea to enhance panoramic 3D object tracking.
>
> C3: Ablation study on multi-camera association to justify claim of "reduces identity switches significantly"
>
> In Table 1 of the paper, we show that our pipeline (detect -> merge -> track) leads to significantly less identity switches (2131 IDS) compared to single-camera (5919 IDS) and detect -> track -> merge (4470 IDS). Thus, our ablation studies indeed support the claims made in the paper.
>
> Li et al., "BEVFormer: Learning Bird's-Eye-View Representation from Multi-Camera Images via Spatiotemporal Transformers", arXiv, 2022.

---

### Meta-Review · Area_Chair_5UFt · 2022-08-15

**Recommendation:** Accept (Poster)
**Confidence:** 4

**Metareview:**

The paper proposes a cross camera three dimensional fusion method for object tracking. The paper is of good technical quality. It is well presented and easy to follow. The reviewer concerns were adequately addressed and it is recommended to include them in the final submission of the manuscript.


**Best Paper Nomination:**

No

---

> ### Author Response · Authors · 2022-08-24
> **Please find attached the updated version of our manuscript and appendix**
>
> Please find attached the updated version of our manuscript and appendix.